# Experiences of COVID-19 infection in North Carolina: A qualitative analysis

**Justine Seidenfeld**[1,2]*, **Anna Tupetz**[1], **Cassandra Fiorino**[3], **Alexander Limkakeng**[1], **Lincoln Silva**[4], **Catherine Staton**[1,4], **Joao R. N. Vissoci**[1,4], **John Purakal**[1,5]

**1** Division of Emergency Medicine, Department of Surgery, Duke University School of Medicine, Durham, NC, United States of America, **2** Center of Innovation to Accelerate Discovery and Practice Transformation (ADAPT), Durham VA Health Care System, Durham, NC, United States of America, **3** Duke University School of Medicine, Durham, NC, United States of America, **4** Duke Global Health Institute, Duke University, Durham, NC, United States of America, **5** Duke-Margolis Center for Health Policy, Duke University, Durham, NC, United States of America

* Justine.seidenfeld@duke.edu

## Abstract

### Background and aim

It has been demonstrated that marginalized populations across the U.S. have suffered a disproportionate burden of the coronavirus disease 2019 (COVID-19) pandemic, illustrating the role that social determinants of health play in health outcomes. To better understand how these vulnerable and high-risk populations have experienced the pandemic, we conducted a qualitative study to better understand their experiences from diagnosis through recovery.

### Methods

We conducted a qualitative study of patients in a North Carolina healthcare system's registry who tested positive for COVID-19 from March 2020 through February 2021, identified from population-dense outbreaks of COVID-19 (hotspots). We conducted semi-structured phone interviews in English or Spanish, based on patient preference, with trained bilingual study personnel. Each interview was evaluated using a combination of deductive and inductive content analysis to determine prevalent themes related to COVID-19 knowledge, diagnosis, disease experience, and long-term impacts.

### Findings

The 10 patients interviewed from our COVID-19 hotspot clusters were of equal distribution by sex, predominantly Black (70%), aged 22–70 years (IQR 45–62 years), and more frequently publicly insured (50% Medicaid/Medicare, vs 30% uninsured, vs 20% private insurance). Major themes identified included prior knowledge of COVID-19 and patient perceptions of their personal risk, the testing process in numerous settings, the process of quarantining at home after a positive diagnosis, the experience of receiving medical care during their illness, and difficulties with long-term recovery.

**Data Availability Statement:** There are ethical restrictions on sharing the present study data publicly because of privacy reasons. Data are

available from the Duke University Health System Institutional Review Board (contact at 919-668-5111) for researchers who meet the criteria for access to confidential data.

**Funding:** Primary investigator - JP Associate investigators - JS, AT, CS, JRNV. This study was funded by a COVID-19 Research Grant from the Emergency Medicine Foundation. https://www.emfoundation.org/grantee/covid19-research-grantees/ The funder did not play any role in study design, data collection and analysis, decision to publish, or preparation of the manuscript. It was expected to present results at the American College of Emergency Physicians 2021 Research Forum meeting.

**Competing interests:** The authors have declared that no competing interests exist.

## Discussion

Our findings suggest areas for targeted interventions to reduce COVID-19 transmission in these high-risk communities, as well as improve the patient experience throughout the COVID-19 illness course.

## Introduction

The coronavirus disease 2019 (COVID-19) pandemic has disproportionately impacted marginalized communities in the United States, especially Black and Hispanic populations [1,2]. These groups are especially vulnerable due to numerous social and socially-mediated medical factors [3–8], and it has been repeatedly demonstrated that they have higher case rates of COVID-19 [5,9], higher rates of hospitalization [10,11], and greater mortality compared to their non-Black and non-Hispanic counterparts [9,12–14]. COVID-19 has clearly illustrated the role that social determinants of health play in health outcomes [15], but the pandemic also poses great danger of further exacerbating these existing inequities [2,16,17].

While there is a large body of epidemiological and other quantitative data to investigate these trends, there has been less research focused on understanding the perspectives of COVID-19 patients, especially those from these vulnerable communities. The numerous challenges to conducting qualitative research in the setting of the COVID-19 pandemic are well known [18,19], yet qualitative work can provide special insights to inform future public health efforts that are specific to the needs of these communities [20]. Previous qualitative work has focused on the experiences of healthcare providers during the pandemic [21–24], as well as various communities who were affected generally by the pandemic and quarantine procedures [25–28]. There is a small and growing body of literature exploring various aspects of the COVID-19 patient experience, including interview studies of hospitalized Latinx [29] and Black [30] patients in the U.S., hospitalized patients in China [31], and COVID-19 "long-haulers" in the U.K [32]. However, there is great need for more insight into how vulnerable and high-risk populations viewed their disease course, the pandemic, and how social risk factors have specifically influenced their experiences.

We used qualitative methods to explore the experiences of diverse COVID-19 patients selected from neighborhoods with disproportionately high rates of COVID-19 infection. This study provides a unique opportunity to better understand the COVID-19 illness experience, from diagnosis through recovery, of those most impacted by the pandemic. We believe that these findings can inform the pandemic response by healthcare systems and public health entities that are tailored towards the most vulnerable, to create community-specific responses to reduce transmission and address both the physical and emotional toll of COVID-19.

## Materials and methods

### Ethics approval

This study was approved by the Duke University Health System Institutional Review Board (IRB# 00105970). Verbal consent was given by all participants. We followed the consolidated criteria for reporting qualitative research (COREQ) guidelines [33].

### Study design

We conducted an exploratory qualitative study, using semi-structured in-depth interviews and an inductive thematic analysis approach [34]. Participant recruitment began in February 2021,

and interviews took place from February 2021 through March 2021. Data analysis continued through September 2021.

## Research team and reflexivity

The research team involved in data collection and data analysis consisted of two emergency medicine physicians (JDP, JS), a registered nurse (AP), a medical student (CF) and a postdoctoral fellow (AT). The postdoctoral fellow with expertise in qualitative research designs trained the team members in qualitative data collection techniques and oversaw the data analysis process. The research team was not actively involved in the previous medical treatment of the study participants and were not known to the participants. AP and CF were bilingual in English and Spanish, and conducted the interviews with our Spanish-speaking participants.

## Setting, participants, and recruitment

The study participants were previous patients above 18 years of age within a North Carolina healthcare system who tested positive for COVID-19 between March 2020 and February 2021. The healthcare system comprises three hospitals and a network of outpatient facilities, with approximately 67,000 yearly admissions and 2.2 million outpatient visits. The patients included in this study received COVID-19 testing at any of the health system's testing sites including Emergency Departments (EDs) and clinics. To recruit participants, we performed a spatial distribution of a total of 13,733 patients who tested positive for COVID-19 to identify population-dense COVID-19 outbreaks (hotspots) through geo-correlation and Kernel Density Estimation. Spatial autocorrelation analysis was performed to identify census tract level clusters of White, Black, and Hispanic populations, based on the 2019 American Community Survey dataset (ACS-5). Patients were identified for recruitment by a randomized sampling method using a computational random number generator (randomizer.org).

## Theoretical framework

This is a qualitative study, following a phenomenologically informed approach, focusing on the experiences of individuals previously infected with COVID-19. We created a semi-structured interview guide. The first section asked about patients' COVID-19 experiences, and a second section was guided by more specific questions related to previously defined social determinants of health [35,36]. The interview guide can be found in S1 File. The interview guide was discussed with the entire study team and piloted with individuals who tested positive for COVID-19, but were not within our healthcare system. The pilot interviews were discussed within the study team and feedback from the pilot interviewees were incorporated to adjust the interview guide. The interview guide was developed in English, then translated and back-translated to Spanish, and validated for accuracy by three members of the study team who were fluent in both English and Spanish.

## Interview procedure/data collection

Eligible participants were approached maximally three times by phone and provided verbal consent prior to the phone interview. We attempted to reach 158 patients identified in the census tract level clusters using available contact information in their health records, and 10 agreed to participate. The interviews lasted between 20 and 50 minutes. The interviews were audio recorded using an encrypted and IRB-approved recording device and stored in a secure university folder that only the study team had access to. The recordings were then transcribed by the interviewers and checked for accuracy by AT. Spanish-speaking interviews were first

transcribed in Spanish and then translated and back-translated following the same translation approach as for the interview guide development. Interviews were conducted in February 2021 and March 2021.

The interviewer's names and roles in the study were provided to the participants prior to the interview, but no prior relationships to the participants were established, and no additional information was shared about the interviewers with the study participants. The interviewers took field notes during the interviews. We did not perform any repeat interviews and did not return the transcripts to the participants for comment and/or correction. To determine information saturation, and therefore our stopping rule to conduct interviews, the interview transcripts and main topics emerging in each interview after the first round were reviewed with the study team before moving forward with data collection. We completed data collection after 10 interviews, similar to a prior review suggesting that the highest levels of saturation using inductive analytic methods are typically achieved with 11–12 interviews for a homogenous sample [37].

### Data analysis

Interviews were analyzed with QSR International's NVivo 12 software using an inductive and deductive thematic content analysis approach. JDP and JS independently both coded two interviews, creating separate codebooks which were subsequently discussed and combined to create a single codebook. The codebook was then discussed with the rest of the research team and changes were made iteratively throughout the coding process and discussed within the team. To ensure coding consistency and avoid coder drift, JS and JDP both coded 30% of the interviews individually, and these were compared by AT. Discrepancies were discussed within the research team until the two coders reached agreements on the coding. Each coded interview was reviewed by AT to further validate the coding procedures.

## Results

The study sample consisted of 10 participants. The sample was equally distributed by sex, with age range of 22–70 years (median 56.5 years, IQR 45–62 years). The majority self-identified as Black (n = 6), and not Hispanic or Latino (n = 9). Participants had dates of first diagnosis of COVID-19 from March-May 2020 (n = 3), June-August 2020 (n = 5), and September-December 2020 (n = 2). Most participants did present to the ED (n = 7) and the majority were not hospitalized (n = 4 of ED patients, n = 7 of total sample). The majority were publicly insured (n = 5) or uninsured (n = 3). See Table 1 for participants' sociodemographic characteristics.

We identified 5 themes and 17 subthemes related to the experiences of the participants before, during, and after their infection with COVID-19. Please refer to Table 2.

### Theme 1: Prior knowledge of COVID-19

**Perceived personal risk of getting COVID-19.** Many participants (n = 6) thought they had minimal to no risk of getting COVID-19. One participant reported that they did not understand how wide-spread COVID-19 would become in their community and believed that early reports were exaggerated:

*I knew about the possibility of [COVID-19], but like everybody else. . . I just never thought that it would hit so much." [P1]*

Many of these participants also reported that by personally following public health recommendations such as using a mask, hand-washing, or following stay-at-home orders, they were mitigating their individual risk and that these actions gave them a sense of safety:

**Table 1. Participant demographics.**

| Characteristics | n = 10 |
|---|---|
| **Age** | |
| 18–30 | 2 |
| 31–60 | 5 |
| 61+ | 3 |
| **Sex** | |
| Male | 5 |
| Female | 5 |
| **Self-identified race/ethnicity** | |
| Black | 6 |
| White | 3 |
| Hispanic Mexican | 1 |
| **Date of COVID-19 Diagnosis** | |
| March-May 2020 | 3 |
| June-August 2020 | 5 |
| September-December 2020 | 2 |
| **Presented to the ED** | |
| Yes | 7 |
| No | 3 |
| **Hospitalized** | |
| Yes | 3 |
| No | 7 |
| **Health Insurance** | |
| Public | 5 |
| Private | 2 |
| Uninsured | 3 |

> *"I felt like I was not going to catch [COVID-19] because I was following instructions, distancing, hand washing, hand sanitizing, wearing a mask. Staying at home. I was doing everything I was supposed to do. . . I wasn't worried about it."* [P2]

**Table 2. Interview themes and subthemes.**

| Theme | Subthemes |
|---|---|
| Prior knowledge of COVID-19 | Perceived personal risk of getting COVID-19<br>Ability to limit exposure or risk<br>Understanding of transmission<br>Sources of information |
| COVID-19 testing process | Reasons for testing<br>Barriers to testing<br>Reaction to a positive COVID-19 diagnosis |
| Ability to isolate during quarantine | Interactions with household members/family<br>Logistical/resource-related aspects of quarantining |
| Experiences with medical care and health systems | Concerns about care provided<br>Positive treatment experiences<br>Issues with access to care because of transportation<br>Contact from the Department of Health<br>History of healthcare inequities influenced behaviors |
| Long term impacts of COVID | Impact on physical health<br>Long-term financial impacts<br>Created anxiety on multiple levels |

However, one participant described inconsistent behaviors prior to formal public health recommendations:

*"In the beginning, I would wear a mask randomly, because at that point we didn't have to wear a mask."* [P3]

When discussing potential exposures that may have been the source of their COVID-19 infection, three participants reported that they were still unsure of how they did contract COVID-19 despite the precautions that they took:

*"And here I am, listening to Fauci, wearing my mask, doing what I'm supposed to do, and I end up with COVID!. . .My family was like, 'What happened to you?' and I said, 'I don't know.' It was just that simple."* [P2]

**Ability to limit exposure or risk.** Additionally, while five participants felt that they were able to use all desired methods to limit their own exposure, four acknowledged that they had risks that might have exposed them to COVID-19. As the following participant who worked for the postal service explained, some of these risks were unavoidable, mostly due to work requirements as essential workers:

*"Going to work. I know I'm at risk being in the mailroom but there is no way for me to work from home and we were always considered essential. People need their mail."* [P4]

*"I continuously work, and worked through all of COVID except for the time that I actually was sick. But the daycare stayed open through it all. Just the transmission and everything, and as essential workers, we had to stay open. And that's something I wish we could have done."* [P3]

Another described the likely possibility of contracting COVID-19 from unavoidable exposure to household family members that were required to work during the pandemic as well.

*"But I caught [COVID-19]- my son came up positive. And so I got myself tested. . . .And I came up positive. He works at a grocery store. And, I guess he caught it from somebody in there or, I caught it at work. I don't know which one of the two, but once he came up positive, I went and got tested."* [P5]

However, two participants acknowledged that some high-risk behaviors of their own which may have exposed them to COVID-19 were voluntary, and expressed a willingness to accept that risk in order to maintain their daily life:

*"If I want to catch the bus, I'm going to catch the bus. Whatever I generally do."* [P6]

Two participants did recognize that their prior health status could impact their personal risk of contracting COVID-19, or having serious symptoms from COVID-19, especially if they had prior existing chronic health issues. However, another participant reported that they were not concerned about catching COVID-19, because they believed they were otherwise healthy:

*"I wasn't concerned. I was healthy. If I got [COVID-19], I figured I got it, and I would deal with it."* [P7]

**Understanding of transmission.**    Most (n = 6) participants described respiratory or droplet transmission as the main mechanism for transmission of COVID-19, and one participant cited contact transmission as well. One participant reported an understanding that older individuals were at greater risk of transmission:

*"I knew that it did not have a super significant impact on younger people, and that it hit a lot harder with older people. Highly transmissible, through the air." [P8]*

**Sources of information.**    Participants reported a variety of sources for information about COVID-19 and recommendations for reducing their risk. Many (n = 8) reported getting information from television news, social media, or online content. One participant reported getting information about COVID-19 from the Department of Health through their workplace. One participant who works in the medical field reported talking to coworkers and relying on prior knowledge of similar viruses. Most of these participants described a combination of sources:

*"Probably a cross section. Yeah, cable news, [I] looked up a little bit online. . .just different resources. I think I did go to the [Centers for Disease Control] website early on and look that up." [P7]*

## Theme 2: COVID-19 testing process

**Reasons for testing.**    Patients were tested in a variety of settings, including community testing sites and the ED. Most (n = 6) patients elected to get tested because of symptoms that they suspected might be COVID-19, ranging from mild symptoms like sore throat, to more severe presentations like significant shortness of breath or fatigue. Given concerns about unnecessary exposure in a healthcare setting, one participant did discuss their symptoms with a medical provider on the phone prior to testing, and was given advice on what circumstances or symptoms would be concerning enough to seek in-person testing and treatment.

*"They told me to keep track of my fevers- it started on a Friday, they told me to quarantine, but my temperatures never came down. My temperatures were 101.4. Even Tylenol didn't help when it got really bad, so my doctors told me to come in and get checked out, and they told me I was positive." [P2]*

Of those with more severe presentations at the time of testing, two patients reported initially starting with mild symptoms that they did not realize could be due to COVID-19, and then became concerned as the symptoms then became more severe.

*"I did not think I had it at all. The first two days, I thought I had a regular cold coming on. . . [Interviewer: What happened that made you decide to get a test?] My symptoms got worse quickly." [P4]*

However, three patients were asymptomatic or very mildly symptomatic at the time of their test. Reasons for testing included work requirements, travel plans, and exposures to family members or other individuals who had tested positive for COVID-19.

**Barriers to testing.**    While the majority felt it was reasonably easy to obtain testing, two patients described barriers to COVID-19 testing. One subject who believes they were ill with COVID-19 early in the pandemic reported that they felt too ill to leave the home for testing,

and subsequently they were never formally diagnosed during this period. This patient then tested positive for COVID-19 about two months later during a workplace screening, at which time they were asymptomatic:

> "When I got first diagnosed, it was in July. Within myself, I'm knowing myself and my body, I probably had it before then, but didn't give it the name or anything like that. I was previously sick, it started one time in May. And I was down for maybe three weeks. I think more or less it was COVID, because it was what I experienced, the majority of the symptoms of COVID. When I said I was down three weeks, I mean exactly that. When I got it in July, I felt nothing." [P3]

Another participant with mild symptoms reported being unable to find community options for rapid COVID-19 testing, and thus chose to go to an ED for testing:

> "I couldn't find any place else [other than the ED] where they were doing [testing]. . . they hadn't informed me where they were doing it and also I wanted the results quickly. The other places required an appointment." [P9]

**Reaction to a positive COVID-19 diagnosis.**    Many patients (n = 7), even when not severely symptomatic at the time of their diagnosis, reported substantial fear or anxiety upon diagnosis, and many shared their worries about dying:

> "When I got the diagnosis, I was fearful. . .I was angry because I felt like I should not have caught it. I should not, I should not. Every time they showed that molecule, I would get mad. I would say, 'This is inside my body. That thing should not be inside of me.' And then I started hearing about all these people dying, and thought, 'Oh my god, this could be me. . .' But it was terrifying to know that I have a terrible disease, and it wasn't curable. That was really terrifying." [P2]

Reasons for these reactions included that so little was known about COVID-19 at the time, and concerns about having already unknowingly exposed others to COVID-19.

> "I kinda was like, 'Why me?'. . .Then I thought about like, you know, wow, I can't really go around family, or anything like that because I didn't want to risk exposing other people. You know, I mean, pretty much I was already trying to do social distancing before this. . . I wasn't around a lot of people as is. Just the whole idea of thinking that I might pass away. . .because I had a few family members of mine that passed away from COVID symptoms." [P10]

Two of the patients who were asymptomatic throughout their COVID-19 experience reported not feeling concerned or as worried even when formally diagnosed, and instead were more focused on the isolation requirements or the other impact it would have on their daily lives:

> "I didn't feel anything. I'm sitting at home doing nothing, my chest not hurting, my head's not hurting, nothing. I still stayed in for 14 days, and my son, at the time, wasn't with me- he was with his father. I was just quarantining in the house by myself, thinking, 'I'm ok.' I just remained inside." [P3]

### Theme 3: Ability to isolate during quarantine

**Interactions with household members/family.** For the seven participants that were discharged home from either the ED or the inpatient hospital and told to stay in quarantine, the majority (n = 5) indicated that they felt mostly successfully able to comply with these directions. However, it often required some amount of daily interaction with at least one household member, usually a spouse or an adult child, in a supportive role to facilitate their compliance:

*"[My husband] stayed at home, but he didn't isolate [himself from me]. He stayed at home. We set up, you know, a system that he will come to the door and check on me, and get my food." [P1]*

For these household members that were providing help to the participants, three also stayed isolated as well during this time with the understanding that the family members that were exposed to COVID-19 needed to isolate as well, even if these household members did not have a formal COVID-19 diagnosis themselves:

*"[My fiance's] a trooper. We live together, so we were both confined for 14 days."[Interviewer: Did she also get sick or test positive?] "Uh, no, she didn't. . . She did not have it at all." [P5]*

Many household members were able to help during their quarantine period without difficulties, however one participant reported that her household partner might have gotten ill during quarantine, but hid this from the participant because of their caretaker role at the time.

*"In retrospect, the thing he actually told me that he felt like he had become symptomatic, but he was afraid to tell me that, because he felt like he had to take care of me. So he was never diagnosed, but we have always felt like he also had COVID." [P1]*

One participant did report that he did not follow the recommendations to quarantine himself, attributing it to difficulties staying at home for so long. This participant lives alone in his home and did not describe having direct household assistance:

*"I did put on a mask and go outside or whatever. No more than 5–10 minutes. I just couldn't stay 10 days in my house. I'm sorry.. . .I went for a check-up and was fine anyways. You know what I mean." [P6]*

Three participants mentioned that they had younger, dependent household members live with other family members at separate residences during their quarantine. For some, this was a source of distress and difficulty, especially for the participant with a newborn at the time of their COVID diagnosis:

*". . .And my kids were with my mother-in-law. I would pump my milk and [my mother-in-law] would come to pick it up at the door. Because of this, my baby was far away from me during this time." [P9]*

**Logistical/Resource-related aspects of quarantining.** Most (n = 8) participants described having enough physical household space and resources, such as access to a bedroom and bathroom that they alone used within their household during their quarantine. It appears that this allowed them to isolate themselves at least somewhat from the household members on whom they were still dependent during their quarantine.

*"Oh, I was able to isolate. . ..We have adequate space to isolate, and that's what we did. . ..we had more than one bathroom. . ." [P1]*

Participants reported a variety of experiences with needing resources such as food and medicine; some reported they had friends and family able to help them as needed. Other participants reported that they had supplies readily available at home, either through prior efforts once they became aware of the pandemic and stay-at-home orders, or via the ability to order food or supplies without direct human interaction.

*"Yes, actually I was home by myself at the time. I really didn't need anything, because when COVID came, I started stocking up on everything." [P3]*

Finally, one participant discussed using telehealth as an alternative to an in-person clinic visit during their quarantine while ill:

*"When I was sick, I had to televisit with my doctors because I couldn't come in[to the office]." [P2]*

### Theme 4: Experiences with medical care and health systems

**Concerns about care provided.** Four participants described negative experiences when asked about the treatment they received for the COVID-19 infection. These experiences ranged from frustration with quality of care to lack of understanding of testing policies and procedures. One participant described being extremely frustrated as she had initially been turned away when she knew she felt too sick to not be admitted.

*"I tried to explain to them what I felt like with the diagnosis of pneumonia with COVID, that I felt like I should stay in the hospital but they were saying that, you know, I could be managed at home. They explained to me that the health department would be following me, calling me every day and if I was—if I was to get worse, I was to come back. You know, the standard kind of thing, but I was pretty angry about it. I was pretty angry about the care, the decision to send me home, yeah." [P1]*

The same participant later returned to the hospital and was subsequently admitted, but then further reported that she was discharged too soon, and attributed it to the increased patient volumes the hospitals encountered during the pandemic.

*"They discharged me. . .still. And that's really—that was their justification for sending me home, and the fact that they needed the bed. And they told me that. That people were coming in left and right, and they were making decisions on priority, on who could go home so they could bring other people in." [P1]*

Another patient expressed anger and frustration at the ED, as they believed their symptoms of shortness of breath were unrelated to COVID-19. The patient goes on to describe a sense of alarm when the COVID-19 test was performed, as he was unaware of the invasiveness of the test itself and its relation to his symptoms.

*"I thought I had a breathing problem. A breathing situation. I went in for, a breathing. . ..a breathing situation and all of a sudden they shot something in my nose [referring to the COVID testing swab] and three minutes later, I got COVID. . ..I didn't go there for no*

*COVID. I didn't go there to take no test, because I didn't know anything about no test until they got my breathing situation right."* [P6]

He also expressed that this experience during his ED visit has changed his feelings about interacting with this particular health care system in general, notably one that he has had a long-standing relationship with.

*"I'm so disgusted with that place. And I went with that place for 18 years of my lifetime. We can carry on this conversation about something else, but I am just letting you know how I really felt about how they told me after the fact that I had COVID and then they blocked the room off and at some point, the room. . ..and everyone looking at me like I'm positive. . ."* [P6]

**Positive treatment experiences.** Half of the participants (n = 5) described overall positive experiences with their testing and treatment. Participants did cite feelings of anxiety or fear, and a sense of professionalism and precaution taken by the ED staff. Though some had feelings of fear of having to go to the ED, the overall sense in these participants was that the emergency department staff was adequately performing their duties under duress.

*"In the emergency department, the care was pretty good. Everybody was taking the necessary precautions with me. Once I got to the wards, it was a little scary. Here they are all by yourself, no visitors, you know? The care was good, I cannot complain about anything. Just the thought of it maybe being your final days, and being on your own was scary."* [P4]

**Issues with access to care because of transportation.** While most participants (n = 7) reported that they had their own vehicle for transportation, one participant reported that he observed that his neighbors would rely on help from others for transportation to their physicians if they couldn't drive themselves, but could still maintain access to their usual health care.

*". . .And the people that I know that basically can't get out, they have people that go with them to the doctors or whatever. They were able to go to their appointments."* [P2]

**Contact from the Department of Health.** Three participants described interactions with the North Carolina Department of Health and Human Services (NCDHHS) or other public employees during their quarantine to monitor or track them. Two felt it was helpful or reassuring, whereas one described it as invasive or stressful:

*"It was very stressful. I got numerous calls from the health department, even though they told me that the health department would be calling daily. There were days when I would get a call from the health department three or four times, which scared me also. They would call from one section of the health department, or their records showed that no one had called me. There were days—there was never a day that nobody called me, it was more days when I would get multiple calls from the health department. And one day, someone showed up at my door from the health department, and I understood what happened with that, but being a nurse for many years, I honestly felt like they were checking to make sure that I was on quarantine—that I was there, and being quarantined."* [P1]

**History of healthcare inequities influenced behaviors.** Two participants noted that the history of healthcare for people of color is likely to have influenced their behaviors during the

pandemic. Additionally, both participants noted that it may influence their decision-making when it comes to a COVID-19 vaccine, which was not yet widely available at the time of this interview.

> *"To be honest with you, I was kind of skeptical about this, the vaccine. I'm not 100% sure if I'm gonna get it or anything like that, because I don't know. . . It's just a lot going on, being that I'm African American, I'm more. . .you, and this is just being honest, a lot of people are skeptical about it, because. . ..I don't know. I can't say, "being targeted," or anything like that, but I'm just not sure." [P10]*

> *"That is a good question, and I feel that; I hear so many people talking the historical situation with the Tuskegee situation. That needs to be clarified to people of color." [P2]*

## Theme 5: Long term impacts of COVID

**Impact on physical health.** All subjects described longer-term symptoms that they attributed to their COVID-19 illness, lasting beyond the acute phase. Three participants reported no lasting physical health impacts beyond a week or two. Six participants described long lasting changes in their breathing or shortness of breath, or persistent fatigue.

> *"My breathing is different. . .Like, I'm more short-winded now. . .It made [going to work] a little bit harder for me because. . .I'm pretty strong for my age. But, you know, I can tell the difference. That's the only thing about it. . .A little short winded now." [P5]*

Of those, one reported that because of lung scarring, she had started a new daily medication to manage her persistent symptoms. Another of those participants expressed frustration at the idea that COVID-19 was still affecting them.

> *"I'm on a rescue inhaler, I'm on one inhaler daily. Because, my pulmonologist says, because of the bilateral pneumonia. She felt like I've had some scarring there. And so I'm on one inhaler a day, and then I have a rescue inhaler." [P1]*

> *"I still don't feel good. I feel weak. That makes me feel bad, too, because I feel like I'm not over it. I feel like this is still in my body. That's the part that's really upsetting me. It hasn't really gone anywhere. And with there not being a cure. . ." [P2]*

One participant described losing taste and smell perception, and another described persistent chest pain for six months after her illness, but these symptoms resolved in both cases.

**Long-term financial impacts.** Four participants discussed substantial stress from the long-term financial impact of the COVID-19 pandemic, typically due to losing time at work or losing a job entirely.

> *"Financially, we're a year behind in a lot of things, because I was working and my pay was dependent—I mean, it took care of our bills. So, I didn't work for a year. . .But in addition to that, in the height of things, we had to readjust, redo our budget. Like so many other people, you know, call people and put on delays on payments for things. . .Take advantage of some of the offers from credit cards, you know, for two or three months that you don't pay, with no interest. We have exhausted all of that." [P1]*

One of these participants reported that he had used his sick leave while he had COVID-19 to receive some salary from his employment, but that he ultimately did lose hours and pay

during his illness. Another was disappointed by the lack of government or workplace solutions for individuals who were impacted by their COVID-19 illness.

*"[I was]out of work for almost a month. [It] affected the money situation, [I] wasn't bringing home as much with short term disability-that thing is horrible. So, I was not bringing as much home, which causes some stress. Also, my work did not have anything set up to help people out, which was frustrating." [P4]*

*"I think the government needs to do more for the people that have had COVID. And they could do more for them. . .A good thing would be to help people when they got COVID, because it puts them behind in their bills. And that's one thing that the government needs to do. . .You know, it really changes your life." [P5]*

An additional participant also linked the problem of financial insecurity to criminal activity in the community:

*"I see that with the crime rate. People out here trying to survive. They [are] out here robbing or shooting and I think that had to do with COVID, too, because people will do anything to provide for their family." [P3]*

**Created substantial anxiety on multiple levels.** Multiple subjects (n = 7) also spoke to the emotional or psychological impact of COVID-19 on their lives, citing increasing anxiety, especially early in the pandemic. One also expressed this on a global level:

*"Of course, you're isolated in a room, so all you're doing is watching television. . .when you could breathe. And what was happening around the United States, and the world. . .so the scare was just unbelievable." [P1]*

Other subjects discussed the direct impact of their COVID-19 diagnosis creating more anxiety within their families or close contacts, many of whom tried to rely on public health measures in response.

*"Family-wise, it made everyone more cautious. We were already doing everything we thought we were supposed to do, but started doing it all even more and just being extra careful, you know?" [P4]*

*"[My family] felt like, if [I] got it, they could get it too because they know how meticulous I am. I had family wearing gloves, masks. . .two masks, everything. They are being very, very careful and taking it seriously. So that's what happened with them." [P2]*

Participants also cited that their COVID-19 illness impacted their own mental health as well, including increasing anxiety about their own or their family's health in general or re-contracting COVID-19, as well as more anxiety about their community's behaviors:

*". . . I don't know if it's the mental part, I'm not going to say I'm more conscious, but now if I see someone without a mask, I'll say something. Because it's like, if you ain't have it, you wouldn't know. What I went through, I don't want anyone to go through what I went through. I don't want children to go through it. I don't want grown people to go through it. It was so much." [P3]*

One participant did cite feeling less anxious as more information became available about the COVID-19 pandemic:

*"Going into last year I was frightened, because they couldn't tell us much. But, now that we are getting more information, it makes me feel better about it. I'm not as frightened as I was before. Even though we don't have a cure yet, I feel better." [P2]*

## Discussion

The goal of this qualitative study was to explore the experiences of COVID-19 patients in a North Carolina health system. This report adds to the small body of literature on patient experiences with COVID-19 testing, illness experience, and recovery. It is novel in that it draws from communities with disproportionately high rates of COVID-19 among population groups of different races and ethnicities, and examines patient experiences with COVID-19 illness outside of the hospital setting. Major themes identified included prior knowledge of COVID-19 and patient perceptions of their personal risk, the testing process in numerous settings, the process of quarantining at home after a positive diagnosis, the experience of receiving medical care during their illness, and difficulties with long-term recovery. These findings have implications for providers, public health entities, and policy makers to improve care for patients in these high-risk communities.

We had several findings that suggest areas for targeted interventions to reduce COVID-19 transmission in these communities. One key finding was that most participants, especially early in the pandemic, believed that they were at low risk of contracting COVID-19. Many reported that they were using public health recommendations like hand-washing, social distancing, and wearing masks. It is known that these actions are protective against COVID-19 and other viral and respiratory illnesses [38,39]. We also know that they are most effective when used together consistently [40], and that there has been some evidence that when patients use one type of public health intervention, it may lead to risk compensation behavior [41]. So, while we cannot truly identify how each of our participants contracted COVID-19, it may be worth further investigations evaluating specific patterns of use of each type of preventative measure in these communities in order to best target interventions that might be needed in communities with high rates of COVID-19 transmission. Additionally, a few participants noted that they were likely exposed through their workplace outside the home or through household members that were essential workers. This is consistent with prior work demonstrating that essential workers within minority groups are disproportionally at higher risk of COVID-19 transmission [42–44], although the specific reasons for their higher risk are perhaps not as straightforward as just purely based on employment outside of the household [45]; future policy work may need to be directed at complex issues of how to better define and protect essential workers [46,47]. Additionally, a few participants also noted that while they started with mild symptoms, it took a more severe course of illness for them to recognize the possibility of COVID-19, seek testing, and appropriately isolate themselves. This delayed diagnosis may have worsened community transmission, similar to concerns about asymptomatic COVID-19 cases [48]. As such, more public health messaging efforts may need to be directed to early symptom recognition. Finally, a number of participants described that while they were isolating at home after a positive diagnosis, they needed daily support from at least one household member who would then be exposed. Household contacts are well established as high-risk [49,50], especially those of essential workers [51]. To reduce transmission in these communities, interventions may need to be targeted directly to caregivers in COVID-19-positive households.

Our findings also demonstrate a number of areas in which COVID-19 patients had significant distress and frustration in their illness course, which would warrant targeted interventions

to improve their patient experience. Many of our participants noted developing substantial fear and anxiety upon receiving a positive diagnosis, consistent with prior work identifying the stress, isolation, and stigma around the COVID-19 illness [52–54]. It may be worthwhile for providers to deliver more support and resources to patients at the time of testing, and to recognize the emotional implications of a positive COVID-19 diagnosis. Multiple participants felt like their care was not well explained to them or they were not taken seriously, and so we suggest that providers are careful to communicate with patients around all aspects of testing and treatment. We also note that a few patients cited inconsistent interactions with the Department of Health as causing more stress or anxiety, so we would recommend more consistent or targeted communications from public health agencies as well. Finally, multiple participants discussed the toll that their COVID-19 illness was taking on their emotional state long-term, with substantial anxiety related to isolation, concern for their families' safety, or interacting with their communities. These findings are aligned with prior work and calls for more efforts to mitigate the so-called "second pandemic" of COVID-19, with great concern for existing behavioral health resources being unable to meet the coming demand in the next few years [55–57]. While these patients receive emotional support from friends, family, and community groups like faith-based organizations, it is incumbent upon the healthcare field to recognize the need to invest in resources to meet these patients' needs [58].

We acknowledge a few limitations in this study. For one, our participant sample was predominantly composed of Black participants, with only one patient from a Hispanic community. However, we believe these qualitative results still provide significant and nuanced data to further inform quantitative studies of infection rates, and future qualitative work could focus on comparisons within these groups. Additionally, these patients were drawn from a single health system registry, and so communities with high rates of COVID-19 transmission that were diagnosed outside of this health system may have been missed. However, this health system cares for a large proportion of the surrounding population, and most hotspots identified were near the area. Finally, interviews in this study were conducted by phone, and we recognize that our discussions may have been more nuanced if conducted face-to-face, and could have perhaps elicited additional themes. However, given the risk of in-person interactions during the COVID-19 pandemic, we believe that this was the best possible choice to protect both patients and interviewers.

## Conclusion

This study contributes significant insight to the experiences of highly vulnerable populations as they navigate COVID-19 illness and its impact on their families and communities. We found themes that suggest areas for interventions to reduce transmission in these high-risk communities and to better support the emotional and mental health needs of patients. It is incumbent upon health systems and public health entities at the federal, state, and local levels to develop tailored tools that can improve the pandemic response for these groups. However, as patients also noted, the health care and research communities have a long history of abuses towards Black and Hispanic populations. The medical community must make concerted efforts to reach out to these communities and earn their trust through transparency in research objectives and methods if these interventions are to succeed.

## Supporting information

**S1 File. English interview guide.**
(DOCX)

## Acknowledgments

We thank Anne McGraw Phillips for her essential help in preparing and reviewing Spanish-language documents.

## Author Contributions

**Conceptualization:** Justine Seidenfeld, Anna Tupetz, Alexander Limkakeng, Catherine Staton, Joao R. N. Vissoci, John Purakal.

**Data curation:** Justine Seidenfeld, Anna Tupetz, Lincoln Silva, John Purakal.

**Formal analysis:** Justine Seidenfeld, Anna Tupetz, Lincoln Silva, Joao R. N. Vissoci, John Purakal.

**Funding acquisition:** Justine Seidenfeld, Anna Tupetz, Alexander Limkakeng, Catherine Staton, Joao R. N. Vissoci, John Purakal.

**Investigation:** Justine Seidenfeld, Anna Tupetz, Cassandra Fiorino, Alexander Limkakeng, Lincoln Silva, Catherine Staton, Joao R. N. Vissoci, John Purakal.

**Methodology:** Justine Seidenfeld, Anna Tupetz, Alexander Limkakeng, Lincoln Silva, Catherine Staton, Joao R. N. Vissoci, John Purakal.

**Supervision:** Alexander Limkakeng, Catherine Staton, Joao R. N. Vissoci, John Purakal.

**Writing – original draft:** Justine Seidenfeld, Anna Tupetz, John Purakal.

**Writing – review & editing:** Justine Seidenfeld, Anna Tupetz, Cassandra Fiorino, Alexander Limkakeng, John Purakal.

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
