## [Decision Letter · Decision Letter 0]

7 Mar 2022

PONE-D-22-03243Experiences of COVID-19 Infection in North Carolina: A Qualitative AnalysisPLOS ONE

Dear Dr. Seidenfeld,

Thank you for submitting your manuscript to PLOS ONE. After careful consideration, we feel that it has merit but does not fully meet PLOS ONE’s publication criteria as it currently stands. Therefore, we invite you to submit a revised version of the manuscript that addresses the points raised during the review process.

We look forward to receiving your revised manuscript.

Kind regards,

Sinan Kardeş, M.D.

Academic Editor

PLOS ONE

Journal Requirements:

Reviewers' comments:

Reviewer's Responses to Questions

**Comments to the Author**

1. Is the manuscript technically sound, and do the data support the conclusions?

Reviewer #1: No

Reviewer #2: Partly

2. Has the statistical analysis been performed appropriately and rigorously? 

Reviewer #1: No

Reviewer #2: N/A

3. Have the authors made all data underlying the findings in their manuscript fully available?

Reviewer #1: Yes

Reviewer #2: No

4. Is the manuscript presented in an intelligible fashion and written in standard English?

Reviewer #1: No

Reviewer #2: Yes

5. Review Comments to the Author

Reviewer #1: Although qualitative study can potentially provide insight into this critical COVID pandemic issue, I would love to see more details coming from higher sample number and fair comparison between different groups. This manuscript has low impact.

Reviewer #2: Summary

This study reports the results of a qualitative interview study of ten North Carolinians who were diagnosed with COVID-19 in the first three waves of the pandemic. The results of the interviews provide new information about individual beliefs during the pandemic and reactions to the health care system both before (e.g. testing), during, and after (long-covid) their illness.

Major items

1. I appreciate that the focus of this study is qualitative, however, it isn’t clear if the conclusions the authors draw from the data are supported by the data since it is unclear how many people are endorsing each of the themes/sub-themes the authors identify. It would be helpful if table 2 was augmented with the number of interviewees who mentioned each sub-theme.

2. Survey issues

a. What was the response rate?

b. Which “census area” did the authors use to identify hot spots?

c. The sample is very different from the population of North Carolina. Can the authors provide a comparison with the demographic characteristics of the identified hot spots?

d. What was the stopping rule for data collection?

6. PLOS authors have the option to publish the peer review history of their article (what does this mean?). If published, this will include your full peer review and any attached files.

Reviewer #1: No

Reviewer #2: No

---

## [Author Response · Author response to Decision Letter 0]

19 Apr 2022

This is additionally included in the "Response to Reviewers" file.

Comment: Reviewer #1: Although qualitative study can potentially provide insight into this critical COVID pandemic issue, I would love to see more details coming from higher sample number and fair comparison between different groups. This manuscript has low impact.

Response: We thank the reviewer for their comment, and appreciate that this is a critical COVID pandemic issue. Our intention with this study was to focus collectively on patients living in high risk COVID-19 hotspot communities as a group that would be high-yield targets for public health interventions and could additionally inform larger studies in the future. We did not intentionally sample from different groups for the purposes of comparison, but this would be an excellent direction for future work, and we thank the reviewer for this idea. We believe that we yielded rich responses from our sample size with 5 predominant themes and 17 sub-themes discussed in this paper, and that we achieved data saturation. This is consistent with prior work finding that 6-7 interviews will capture the majority of data using inductive analytic methods, and that 11-12 interviews is consistent with the very highest levels of saturation [1]. We have added a statement to the methods section on “Interview Procedure/data collection” on page 8 to address this. Please additionally see the response to the reviewer comment on a stopping rule for data collection below as it additionally informs our sample number. 

Comment: Reviewer #2: This study reports the results of a qualitative interview study of ten North Carolinians who were diagnosed with COVID-19 in the first three waves of the pandemic. The results of the interviews provide new information about individual beliefs during the pandemic and reactions to the health care system both before (e.g. testing), during, and after (long-covid) their illness.

Reviewer: We greatly appreciate these kind comments recognizing the novel data in this manuscript. 

Comment: Major items - I appreciate that the focus of this study is qualitative, however, it isn’t clear if the conclusions the authors draw from the data are supported by the data since it is unclear how many people are endorsing each of the themes/sub-themes the authors identify. It would be helpful if table 2 was augmented with the number of interviewees who mentioned each sub-theme.

Response: We thank the reviewer for this excellent suggestion. We have added numbers throughout the text of the manuscript to establish the numbers of patients who discussed each of the themes and subthemes included. We elected to use the text of the manuscript as opposed to the table, as it allows for more detail in each subtheme, but we believe that this addresses the reviewer comment. 

Survey issues

Comment: What was the response rate?

Response: We appreciate the opportunity to clarify this point. 158 eligible participants were identified in the hotspot clusters, and using available contact information in their health records, 10 patients agreed to participate. We have included a statement with this in the methods section on “Interview Procedure/Data Collection” on page 7. 

Comment: Which “census area” did the authors use to identify hot spots?

Response: Thank you for this question, we used the census tract level. We have changed the language in the methods section on “Setting, Participants, and Recruitment” on page 6 to reflect this. 

Comment: The sample is very different from the population of North Carolina. Can the authors provide a comparison with the demographic characteristics of the identified hot spots?

Response: Thank you for the chance to clarify this point. Our randomized sample was taken equally from each of the census tract level cluster of White, Black, and Hispanic populations, as our intention was to collect data from patients of all groups living in hotspot communities, although this would not correspond to the overall demographics of North Carolina. We acknowledge that our patient sample (self-identified as 7 Black, 3 White, and 1 Hispanic Mexican) is not equally representative of all these groups, and future studies could specifically highlight each group. We have added a statement to the limitations section on page 31 to acknowledge this. 

Comment: What was the stopping rule for data collection?

Response: We appreciate this helpful question to strengthen this manuscript. Our stopping rule was based on reaching information saturation, which we defined as when few or no new concepts emerged in subsequent data collection through our inductive content analysis approach. At this point, we reached a nuanced understanding of the perspectives of participants living in our defined COVID-19 hotspot communities. We have added a statement to clarify this in the methods section on “Interview procedure/data collection” on page 8.

---

## [Decision Letter · Decision Letter 1]

19 May 2022

Experiences of COVID-19 Infection in North Carolina: A Qualitative Analysis

PONE-D-22-03243R1

Dear Dr. Seidenfeld,

We’re pleased to inform you that your manuscript has been judged scientifically suitable for publication and will be formally accepted for publication once it meets all outstanding technical requirements.

Kind regards,

Sinan Kardeş, M.D.

Academic Editor

PLOS ONE

Additional Editor Comments (optional):

Reviewers' comments:

Reviewer's Responses to Questions

**Comments to the Author**

1. If the authors have adequately addressed your comments raised in a previous round of review and you feel that this manuscript is now acceptable for publication, you may indicate that here to bypass the “Comments to the Author” section, enter your conflict of interest statement in the “Confidential to Editor” section, and submit your "Accept" recommendation.

Reviewer #1: All comments have been addressed

Reviewer #2: All comments have been addressed

2. Is the manuscript technically sound, and do the data support the conclusions?

Reviewer #1: Yes

Reviewer #2: Yes

3. Has the statistical analysis been performed appropriately and rigorously? 

Reviewer #1: Yes

Reviewer #2: N/A

4. Have the authors made all data underlying the findings in their manuscript fully available?

Reviewer #1: Yes

Reviewer #2: Yes

5. Is the manuscript presented in an intelligible fashion and written in standard English?

Reviewer #1: Yes

Reviewer #2: Yes

6. Review Comments to the Author

Reviewer #1: Authors provided responses and revised their manuscripts to meet expectation. I believe it is acceptable for publication.

Reviewer #2: (No Response)

7. PLOS authors have the option to publish the peer review history of their article (what does this mean?). If published, this will include your full peer review and any attached files.

Reviewer #1: No

Reviewer #2: No

---

## [Editor Report · Acceptance letter]

24 May 2022

PONE-D-22-03243R1 

Experiences of COVID-19 Infection in North Carolina: A Qualitative Analysis 

Dear Dr. Seidenfeld:

I'm pleased to inform you that your manuscript has been deemed suitable for publication in PLOS ONE. Congratulations! Your manuscript is now with our production department. 

Kind regards, 

on behalf of

Dr. Sinan Kardeş 

Academic Editor

PLOS ONE